# Pseudo-Desublimation of AdBlue Microdroplets through Selective Catalytic Reduction System Microchannels and Surfaces

**DOI:** 10.3390/mi14091807

**Published:** 2023-09-21

**Authors:** Claudiu Marian Picus, Ioan Mihai, Cornel Suciu

**Affiliations:** Faculty of Mechanical Engineering, Automotive and Robotics, Stefan cel Mare University, 720229 Suceava, Romania; claudiu.picus@usm.ro

**Keywords:** microchannels, pseudo-desublimation, SCR, AdBlue, NOx

## Abstract

In the present paper, the occurrence and development of the pseudo-desublimation process of AdBlue microdroplets in the microchannels and surfaces of catalytic reduction systems (SCR) are reported. In order to understand how the pseudo-desublimation process develops, the influence of heat flux values on the heat transfer of AdBlue injection was analysed, taking into account the structure of the microchannels inside the SCR and the overall configuration of the installation. The evolution of the AdBlue vapour flow in the SCR system was simulated, as well as the temperature variation along an SCR microchannel through which the mixture flows. An experimental set-up was designed in order to visualise and interpret the processes at the onset of pseudo-desublimation. The results described in this paper confirm the existence of a pseudo-desublimation process that occurs only under certain temperature conditions when AdBlue is injected into SCR systems. The characteristics of the crystals formed and their growth rate depend on the working temperature, which could be controlled by efficient preheating methods immediately after engine start. A better understanding of the process will allow the development of methods of avoiding solid depositions on SCR system components, which has a direct impact on SCR catalyst performance and durability.

## 1. Introduction

The Euro 7 standards for cars, proposed to come into effect in 2025, include a number of stricter emission regulations. Although the new values of NOx (nitrogen oxides) emission limits are not much lower than in the Euro 6d standard, the cold emissions test will mean big increases in costs, mainly for diesel engines. At present, coding to Euro 6d regulations, petrol engines have a limit of 60 mg/km for NOx emissions, while diesel engines have a limit of 80 mg/km. With the Euro 7 rules, the NOx emission limit will be 60 mg/km for all types of internal combustion engines. There is a one-off derogation from 60 mg/km to 75 mg/km NOx for small commercial vehicles up to 3.5 tonnes with a specific power output of up to 35 kW (48 hp). Emissions will be measured using the same method as in Euro 6d, called RDE (Real Driving Emissions), which has been found to be viable and conclusive. The European Commission’s communication does not explicitly mention the introduction of a short-distance cold emissions test for small commercial vehicles with a whole-trip specific power of 10 km. Basically, the kilometre limit will be identical to the normal long-distance test, but will be very difficult to achieve [1,2]. Other changes brought about by the Euro 7 rules include extending the period of compliance of vehicles with emission standards from 100,000 km and 5 years, as was the case under Euro 6d rules, to 200,000 km and 10 years, as the average age of cars in Europe approaches 12 years [1,2].

The amount of pollutants according to European requirements for engines up to 35 Kw diesel and petrol are shown in Table 1, comparatively for the Euro 6d and Euro 7 standards, where PM represents the accepted particulates or atmospheric particulate matter and PN is the particle number (symbol N), which is the number of constituent particles in the analysed thermodynamic system. The carbon monoxide (CO) emissions are currently limited to 1000 mg/km for diesel and 500 mg/km for petrol engines, while the Euro 7 standard will limit CO emissions to 500 mg/km for both types of engines. The total hydrocarbon (THC) emissions are also limited at 100 mg/km. Non-methane hydrocarbons (NMHCs) are also limited for all engines at 69 mg/km. To meet the European requirements shown in Table 1, a selective catalytic reduction (SCR) system is implemented on the exhaust manifold. Selective catalytic reduction (SCR) is a widely used technology to reduce NOx (nitrogen oxide) emissions from engines in both light- and heavy-duty vehicles. It involves the use of a catalyst, typically made of various ceramic materials, with an active catalytic component such as vanadium. In the SCR process, a urea solution is injected upstream of the catalyst.

In the European Union, this solution is known as AdBlue, while in the United States it is called Diesel Exhaust Fluid (DEF). The urea solution plays a crucial role in initiating the catalytic reaction within the catalyst. For catalytic conversion to be effective, the water content of the AdBlue solution must be evaporated and the NH_3_ (ammonia) molecules must be mixed with the exhaust gas. It is important to achieve an even distribution of NH_3_ on the catalyst surface to maximise the efficiency of catalytic conversion. If there is a high local concentration of ammonia, an excessive ammonia emission may result after the process taking place in the SCR. On the other hand, a low local concentration of ammonia may lead to poor NOx conversion. In the Euro 6 emission standard, the permissible limit for excess ammonia in the catalyst has been regulated to ten particles per million (ppm) [3]. This stricter limit has increased the complexity of dosing and mixing urea. To achieve both a high conversion efficiency and lower ammonia emissions, a more advanced and controlled mixing process is required. Urea dosing and mixing technologies aim to ensure an efficient distribution of NH_3_ molecules on the catalyst surface, minimising excess ammonia and achieving optimal catalytic conversion of NOx emissions. These developments contribute to reducing the environmental impact of vehicles and improving air quality by reducing harmful NOx emissions [4,5].

The present study reports how the evaporation and vaporisation of AdBlue droplets in the microchannels of an SCR system takes place in a pseudo-desublimation process. The pseudo-desublimation of AdBlue microdroplets through microchannels of selective catalytic reduction systems is a process that starts by converting the liquid AdBlue solution into microdroplets by injection. Once atomised, due to the temperature in the SCR system, these microdroplets are transformed into aerosols which enter the SCR microchannels as vapour. In the microchannel, depending on temperature, the AdBlue vapour undergoes a desublimation process, which consists of a quasi-instantaneous transformation of the microdroplets into vapour or, at high temperatures, directly into solids. This process occurs due to the small size of the microchannels, their various shapes, as shown in Figure 1, and the velocity of the fluid flowing through it [6,7,8].

The role of AdBlue injection is to disperse the solution in micrometre-sized aerosols in order to achieve a more uniform distribution in the exhaust gas, which contributes to improving the efficiency of the catalytic reduction process [9].

Various studies, mostly theoretical, regarding micro- and nanochannel shapes and their influence on ion transfer phenomena can be found in literature [10,11,12]. The selective catalytic reduction of NOx with ammonia was modelled in previous studies found in the literature [13]. As can be seen from Figure 1, the shape of the microchannels used in the investigated SCR systems is that of a honeycomb, but in the practice of making SCR systems other shapes of microchannels can be used, such as cylindrical, triangular, square, rectangular or complex shapes. Previous theoretical models [13] showed that a reduction in the number of microchannel cross-section sides leads to an increase in the reaction rate due to an increase in the area to volume ratio. However, the results indicated that the rate of reaction is strongly affected by diffusion at the corners of noncircular channels.

Microchannels are used in SCR practice in order to increase the contact area between the exhaust gas of diesel engines and the catalyst surfaces. SCR systems are designed to create an environment conducive to chemical reactions that contribute to the reduction in pollutant emissions, with nitrogen oxides (NOx) being targeted. At the same time, microchannels can also be used to distribute the AdBlue solution evenly in the exhaust gas, thus contributing to the efficient distribution of the reactant throughout the SCR system [14,15,16,17]. Inside these microchannels, the occurrence of a pseudo-desublimation process is undesirable, as it results in solid deposits that ultimately clog the system (see Figure 2).

## 2. Heat Exchange in Microchannels and Other SCR Components

In a first step, the conditions under which the desublimation process occurs as a function of the change in exhaust gas temperature and temperature evolution in certain SCR components were investigated. Literature such as [11,13,18] investigated different geometries and heat transfer aspects in such systems. In the study of heat transfer of an SCR catalyst, a circular microchannel geometry was adopted. In the case of an SCR catalyst, heat transfer takes place between the walls of the microchannels and the AdBlue vapours that are found diffused in the exhaust gas through several mechanisms [18], as shown in Figure 3.

The abovementioned mechanisms include the following:Convection from/to the AdBlue gas–vapour mixture flowing between the microchannel walls: heat is transferred by convection between the mixture flowing inside the channels and the microchannel walls. Typically, heated exhaust gases transfer heat to the walls, and cooled exhaust gases take heat from the walls in the case of the pseudo-desublimation of AdBlue.Radiation from/to the AdBlue gas–vapour mixture flowing inside the channels: thermal radiation can also contribute to heat transfer between the monolith walls and the gas flowing inside the channels. This means that thermal energy is transferred by radiation between the catalyst surface and the AdBlue gas–vapour mixture from the exhaust.Convection from/to the environment: heat can be transferred by convection between the monolith walls and the environment. For example, in a vehicle-mounted SCR catalyst system, the surrounding environment (air) can pick up heat from the catalyst by convection.Radiation from/to the environment: radiative heat transfer between the monolith walls and the environment. This means that heat energy is transferred by radiation between the surface of the catalyst and the surrounding environment.

The abovementioned heat transfer mechanisms are important in evaluating and optimising the performance of SCR catalysts, as the distribution and efficiency of heat transfer can influence chemical reactions and the performance of the SCR system in reducing NOx emissions.

To study the processes in SCR systems, the following variables were considered for the analysis of pseudo-desublimation of AdBlue droplets in a cylindrical microchannel: microchannel diameter D (in m), microchannel length L (in m), AdBlue solution velocity u (in m/s), exhaust gas velocity v (in m/s) and inlet AdBlue solution concentration $C_{in}$.

By applying the mass and momentum conservation equations, partial differential equations describing the pseudo-desublimation process in the cylindrical microchannel are obtained. These equations were adopted according to the models developed by [19,20,21] and can be expressed as follows:(1)$$\frac{\partial C}{\partial t}+u\frac{\partial C}{\partial x}=0.$$

Equation (1) represents the mass conservation law in a microchannel, where C is the concentration of the AdBlue solution, t is the time, x is the axial position in the microchannel and u is the velocity of the AdBlue solution. The equation indicates that the concentration variation as a function of time and position is compensated by the mass transport given by the velocity of the AdBlue solution. The momentum conservation equation in a micro-channel is given by
(2)$$\rho \frac{\partial v}{\partial t}+\rho \cdot u\frac{\partial v}{\partial x}=-\frac{\partial p}{\partial x},$$
where v is the exhaust gas velocity, p is the pressure and ρ is the density. The equation indicates that the variation in momentum as a function of time and position is compensated by the momentum transfer given by the pressure difference in the axial direction and the fluid density [22].

To simplify the system of equations, we will appreciate the following assumptions that the exhaust gas velocity is much higher than the velocity of the AdBlue solution, so we can neglect the terms containing u in the momentum conservation equation.

With this assumption, the system of equations can be reduced to:(3)$$\frac{\partial C}{\partial t}+u\frac{\partial C}{\partial x}=0;\;\frac{\partial p}{\partial x}=-\rho \frac{v^{2}}{D}.$$

The solution of this system of equations depends on the initial and boundary conditions. To determine the concentration distribution of the AdBlue solution in the microchannel, the initial concentration $C_{in}$ and the boundary conditions (e.g., the concentration at the microchannel outlet) must be known. The convection–diffusion heat transfer equation or convection–conduction equation is as follows:(4)$$\frac{\partial T}{\partial t}+v\frac{\partial T}{\partial x}=\alpha \frac{\partial^{2}T}{\partial x^{2}}.$$

Equation (4) describes the temperature variation T as a function of time t and axial position x, given the fluid velocity v and thermal diffusion coefficient α [23].

The calculation of the convective transfer coefficient of the exhaust gas $h_{i}$ upon impact with the SCR catalyst [24] can be performed with the following equation:(5)$$h_{i}=\alpha_{p}\cdot d\cdot L{\left(\frac{\mu }{\mathrm{Pr}}\right)}^{0.33}$$

In Equation (5), α_p_ is a proportionality or correction factor that takes into account the heat transfer at the catalyst surface, d is the catalyst honeycomb diameter, L is the catalyst length, μ is the dynamic viscosity of the exhaust gas and Pr represents the Prandtl number of the exhaust gas [22,23,25].
(6)$$Pr=\mu \frac{C_{p}}{k}.$$

In Equation (6), C_p_ represents the specific heat capacity of the exhaust gas, k represents the thermal conductivity of the catalyst and μ is the dynamic viscosity of the exhaust gas.

The proportionality factor $\alpha_{p}$ will be
(7)$$\alpha_{p}=\frac{h_{gas}\cdot k}{h_{gas}+k}.$$

In Equation (7), h_gas_ is the heat transfer coefficient of the gas at the catalyst surface and k represents the thermal conductivity of the catalyst [26,27].
(8)$$\frac{1}{U}=\frac{1}{h_{\mathrm{gas}}}+\frac{\Delta x}{k}+\frac{1}{{\;h}_{\mathrm{cat}}},$$

The notations in the overall heat transfer coefficient, U, Equation (8), have the following meanings: h_gas_ is the exhaust gas heat transfer coefficient, h_cat_ is the catalyst heat transfer coefficient and Δx is the catalyst thickness.

In order to determine the heat transfer coefficient between the combustion gas and the catalyst, we can use the general formula [28]
(9)$$dq_{conv}=h\cdot dA_{s}\left(T_{gas}-T_{cat}\right),$$
where dq_conv_ is the convective heat flux (in W/m^2^), h is the heat transfer coefficient (in W/m^2^K), A_s_ is the heat transfer area (in m^2^), T_gas_ is the exhaust gas temperature (in K) and T_cat_ is the catalyst temperature (in K) [29]:

The SCR parameters taken into account in the calculations are shown in Table 2.

By replacing the above-described equations in the energy balance, the following equality is obtained:(10)$$\frac{d}{dx}\left(A_{s}\frac{dT}{dx}\right)-\frac{h}{k}\frac{dA}{dx}\left(T_{gas}-T_{cat}\right)=0$$

The global heat transfer rate q_g_ is given by the relation
(11)$$q_{g}=mC_{p}{}_{cat}\left(T_{gas}-T_{cat}\;\right),$$
where C_pcat_ represents the thermal capacity of the catalyst.

The average temperature over the entire length of the microchannel at different temperatures and exhaust gas mass m is determined with the relation
(12)$$T_{b}=T_{cat}-(T_{cat}-T_{gas})\exp\left(\frac{-h\pi DL}{mC_{pcat}}\right),$$
where D represents the catalyst microchannel diameter and L is the total considered length.

The temperature variation various gas temperature values (marked in the graph legend) determined by aid of Equation (11) are graphically represented for various temperature levels, as shown in Figure 4.

The global thermal phenomena for the heat transfer to catalysts with circular microchannels are governed in the case of convection, radiation and conduction by the following equations [23,30].

Heat loss (q) from the SCR catalyst occurs by convection to the ambient temperature (q_conv_) (Equation (13)) and by radiation exchange with its walls (q_rad_) (Equation (14)).
(13)$$q_{conv}=h_{gas}\pi DL\left(T_{cat}-T_{amb}\right)$$
(14)$$q_{rad}=\varepsilon \pi DL\sigma \left(T_{\mathrm{cat}}^{4}-T_{\mathrm{amb}}^{4}\right),$$

Therefore, as the total heat loss is $q=q_{conv}+q_{rad}$ and with A = πD, where A represents the surface area of the pipe, it results that
(15)$$q=h_{gas}AL\left(T_{cat}-T_{amb}\right)+\varepsilon AL\sigma \left(T_{cat}^{4}-T_{amb}^{4}\right),$$
where ε is the emissivity of the SCR catalyst and T_amb_ is the ambient temperature around the SCR catalyst.

Figure 5 shows the variation in the heat transfer, depending on the temperature change of the AdBlue gas–vapour mixture, considering the different heat transfer modes inside the SCR.

We assume that pseudo-desublimation takes place in a cylindrical microchannel of diameter D and length L, through which AdBlue droplets flow at a constant velocity u. After heating, the AdBlue droplets are dispersed as aerosols in the exhaust gas flowing through the microchannel at a constant velocity v in the axial direction of the microchannel. The Nusselt number (Nu) as a function of the friction factor (f), the Reynolds number (Re) and the Prandtl number (Pr) are used to calculate the convective heat exchange in microchannels, taking into account the roughness of the microchannel. If the Nusselt number allows the appreciation of the convective phenomenon, the friction factor takes into account the pressure losses caused by friction in the fluid flow [21,31,32,33,34]:(16)Nuf=f8×Re−100× Pr1+12.7 ×f812×Pr23−1,

Using a calculation code developed under Mathcad, the variation in the Nusselt number with the friction coefficient f was calculated for the flow of the AdBlue exhaust gas–vapour mixture through the SCR microchannels, as shown in Figure 6.

By analysing the obtained results, it can be seen that the Nusselt number reaches a maximum value of 2 × 10^4^ for the maximum value of the friction coefficient.

Figure 7 shows the results of the temperature variation in the catalyst microchannels in the transient regime considering that they have a diameter of 0.8 mm and a length of 200 mm. In order to accurately study the temperature variation, the x-distance of the mixture flowing through the microchannels was also taken into account.

For the transient calculation of the temperature variation along the SCR catalyst traversed by a high-temperature mixture, the relation given in Equation (17) was used.
(17)$$T\left(x,t\right)=T_{in}+\left(T_{out}-T_{in}\right)erf\left(\frac{x}{2sqrt\left(\alpha t\right)}\right).$$

In Equation (17), the parameters T_in_ and T_out_ represent the temperatures (in K) at the inlet and outlet of the catalyst, respectively. The thermal diffusivity coefficient is α (in m^2^/s), and the “erf” function is the Gaussian error function.

Figure 7 shows the results of the calculations obtained by implementing Equation (17) in Mathcad V14.0. The abscissa of the graph represents the distance travelled by the mixture of hot exhaust gas and AdBlue vapour. The multiple plots in Figure 7 represent the temperature evolution along the travelled distance, with increasing of time. Time was considered between 0 and 0.5 s in 0.1 s increments, showing that the SCR temperature increases with the distance travelled but also depends on time.

In a previous study [35], the temperature variation in the wall where pseudo-desublimation occurs was analysed as a function of time and microchannel wall thickness, given the concentration of AdBlue when the rate of heat flow passes through it.

Similar research on the vaporisation of liquid droplets in a high-temperature gaseous environment revealed that the variation in wall temperature is directly dependent on the wall thickness and AdBlue concentration. It was also observed that the temperature increased with time, but the rate of this increase was influenced by the microchannel wall thickness and AdBlue concentration.

In conclusion, the study conducted by Mihai et al. [35] highlighted the importance of controlling variables such as wall thickness and heat flow rate in order to optimise the pseudo-desublimation process and obtain the most accurate results. To determine the microchannel wall temperature T_p_, dependent on time (t) and a specific depth or position $z_{1}$ and taking into account the space travelled by the mixture, the following relation was used:(18)$$Tp\left(t,z_{1}\right)=\dot{q}\frac{\delta_{T}\left(t\right)}{2k}{\left(1-\frac{z_{1}}{\delta_{T}\left(t\right)}\right)}^{2},$$
where $\dot{q}$ represents the density of thermal flow (internal source of generating heat), $\delta_{T}\left(t\right)$ is the distance travelled along the $z_{1}$ direction, through the microchannel, k is the specific constant of the chemical reaction, dependent on temperature, as shown in Equation (19), and C represents the AdBlue vapour concentration.

The results obtained in Mathcad for a specific density of the thermal flow value of $6\times 10^{-3}\;W$ are shown in Figure 8, and it can be seen that the temperature increases with time depending on the thickness of the microchannels, whose values are shown in the figure legend (in millimetres).

Another concern was the analysis of the exhaust gas conversion rate as a function of temperature and microchannel diameter. Since temperature and microchannel diameter influence mass and heat transfer due to the increased contact area between the gas and the catalyst walls, a significant impact on the conversion process of gaseous pollutants occurs.

To determine the conversion rate, a Matlab code was used using the Arrhenius equations. Using these equations, the specific constant of the chemical reaction $k_{a}$ as a function of temperature was calculated:(19)$$k_{a}=k_{0}exp\left(\frac{-E_{a}}{RT\left(j\right)}\right),$$
where $k_{0}$ is a pre-exponential constant, $E_{a}$ is the activation energy, R is the ideal gas constant and $T\left(j\right)$ is the temperature.

The conversion rate $C\left(j\right)$ was calculated using the formula given in Equation (20).
(20)$$C\left(j\right)=\frac{k_{a}\;A\;L\;P}{R\;T\left(j\right)}$$
where A is the microchannel area, L is the microchannel length, P is the pressure and j is the index of the allocation of different temperatures of the mixture. In the calculation, different microchannel diameters were considered: values between 0.1 mm and 0.6 mm for a microchannel length of 200 mm. This process allowed obtaining the conversion rate depending on the specified parameters according to models given in the literature [24,36,37,38].

The graph in Figure 9 represents the conversion rate as a function of the gas inlet temperature through the microchannel for different diameter sizes. Diameter values can be found in the figure legend.

The analysis of this graph (Figure 9) allows observing that the conversion rate evolves exponentially with increasing temperature of the mixture through the microchannel diameter.

A study was carried out taking into account different values of the activation energy ($E_{a}$). For example, at an activation energy of 50,000 J/mol and a microchannel diameter of 0.10 mm, a lower conversion rate was obtained than for an activation energy of 120,000 J/mol and a diameter of 0.30 mm. This result indicates the existence of a dependence between the two variables, showing that microchannel size and activation energy significantly influence the conversion rate.

The concentrations of AdBlue aerosols and their evolution along the microchannel as a function of time were considered, and it was found that their concentration decreases over time due to the pseudo-desublimation phenomenon. This decrease in concentration can be explained by the rapid transformation of the AdBlue solution into a fine aerosol in the microchannel upstream and their dispersion in the exhaust gas. Therefore, as the aerosols move along the microchannel, their concentration gradually decreases.

These results demonstrate the importance of considering the influence of variables such as microchannel diameter, activation energy and aerosol concentration in the catalytic reduction process and in the mathematical modelling of this process. Understanding and controlling these variables can help to optimise the performance of catalytic reduction systems and improve the efficiency of the gaseous pollutant conversion process.

An increase in the activation energy value will lead to a decrease in the NOx conversion rate at lower temperatures. In other words, the higher the activation energy $E_{a}$, the more the chemical reaction requires important temperatures in order to take place to a significant extent. This shows that $E_{a}$ is a key factor in influencing the rate of NOx conversion.

Figure 10 shows how the AdBlue vapour concentration $\left(C_{vap}\right)$ varies with microchannel length (L) and time (t), obtained in Matlab based on Equation (21).
(21)$$C_{vap}\left(t,L\right)=C_{0}\cdot \exp\left(-\frac{K\cdot t}{L}\right),$$
where C_0_ is the initial AdBlue vapour concentration and K is the evaporation rate. This formula describes how the concentration of AdBlue vapour decreases over time as a function of microchannel length and evaporation rate. The higher the evaporation rate K, the faster the AdBlue evaporates, and the faster its concentration decreases over time.

The change in aerosol concentration shows a sharp drop in the initial value in the first 50 s. This indicates that the pseudo-desublimation process has a rapid evolution in the initial stages and generates a significant amount of aerosols in a relatively short time. Studying and modelling this variation in aerosol concentration is important to understand the dynamics and behaviour of aerosols in the catalytic reduction system. By obtaining detailed information about the aerosol concentration variation, the process can be optimised and measures can be taken to ensure a uniform and efficient aerosol distribution in the exhaust gas.

## 3. Modelling of AdBlue Vapour Mass Change in the Pseudo-Desublimation Process

Simulations carried out in Ansys Release 2023 R1 for the mass exchange analysis and study of the volume fractions of liquid AdBlue coming into contact with the exhaust gas considered an initial temperature of 25 °C for the liquid solution and a temperature of 500 °C for the exhaust gas.

Figure 11 shows the simulation results for the turbulent kinetic energy, obtained by aid of Ansys, for an SCR system subjected to a process of AdBlue injection in volume (given by the exhaust gas) and on surfaces (walls of the SCR upstream tubing).

From Figure 11, it can be seen that at time t(6 s), the injection of the AdBlue solution takes place, which causes a flow of the droplet jet forming a ring that evolves partially in volume (first part) and then on the surfaces (reaches the pipe wall). At time t(7 s), a rapid diffusion and evaporation of the aerosols can be observed; in fact, the pseudo-desublimation process takes place. The pseudo-desublimation process consists of the direct transformation of a liquid directly into a solid, but unlike pure desublimation, in this case there is an intermediate phase. The intermediate phase consists of the appearance of a vapour phase (between the liquid and solid state) which proceeds so rapidly that it does not remain entirely between the liquid and solid state. Particularly important for the occurrence of pseudo-desublimation is the working temperature. It can be seen from the figure that part of the AdBlue vapour is integrated and entrained in the combustion gases. At time t(14 s), it can be seen from the simulation how the AdBlue vapour–combustion gas mixture propagates through the entire volume of the SCR catalyst.

At time t(18 s), diffusion is observed along the entire length of the catalyst, and most likely under certain conditions even within the microchannels a pseudo-desublimation phenomenon occurs. The statement is based on the experimental finding that solid deposits of the resulting AdBlue salts occur as a result of the pseudo-desublimation process taking place at well-specified high temperatures. The simulations allow us to study the behaviour of the mixture inside an SCR system with the visualisation of the vapour mass exchange and its flow behaviour. The temperature variations in the SCR system contribute fundamentally to the occurrence or non-occurrence of the pseudo-desublimation process (a totally undesirable phenomenon) so that in operation it can be avoided by using the appropriate programming environment specific to the gas cleaning system during the use of AdBlue.

Figure 12 shows the temperature variation in a microchannel of a honeycomb in an SCR catalyst with a diameter of 0.8 mm, obtained by simulation in Ansys.

In this representation, a time interval up to t(17 s) is considered, in which the operating regime is transient. AdBlue injection takes place at t(8 s), as illustrated in Figure 12.

The first image corresponds to the initial time t(0 s), which is associated with the period when the exhaust gas mixture encounters the microchannels of the SCR system (SCR cartridge). It can be seen that the velocity of the mixture oscillates when it enters the catalyst (indicated by the detail). In the first two sections, only combustion gases are introduced into the SCR, and between t(4 s) and t(6 s) a gradual temperature increase occurs. After this point, the injection of AdBlue takes place at time t(8 s), when the exhaust gas undergoes cooling. The rest of the sections in the figure show how the temperature changes over time, which can be analysed on the basis of the gradient in the legend.

An overview of the temperature change obtained in Ansys for the SCR assembly is shown in Figure 13, where it can be seen that we have a different temperature range depending on each area of the catalyst.

The microchannel temperature analysis is shown at the top, with maximum values obtained centrally on the SCR cartridge and mixture velocity vectors at the bottom.

## 4. Experimental Set-Up

In the study of AdBlue injection processes in an SCR system, an experimental set-up was designed and built to allow laboratory tests with the aim of demonstrating that under certain conditions the phenomenon of pseudo-desublimation can occur in the case of contact with a strongly heated surface. The main components of the set-up are shown in Figure 14. The experimental set-up allows spraying a controlled jet of AdBlue droplets onto a heated surface in order to study the pseudo-desublimation process.

The meanings of the numbered items in Figure 14 are as follows: 1, AdBlue tank; 2, pressure gauge; 3, tap; 4, AdBlue pump; 5, test surface; 6, electronic module; 7, inductive heating system; 8, hood; 9, fan; 10, Peltier vapour cooling module.

The AdBlue solution is taken from tank 1 after opening tap 3 and sent via pump 4 to an injector (not shown). The AdBlue injector is controlled by control module 6, which can adjust the time between two injections and the amount of solution injected. Pressure gauge 2 measures the pressure in the system, generated by pump 4. The jet of AdBlue microdroplets reaches test surface 5, which can reach temperatures ranging from ambient to a maximum of 600 °C. Heating to any temperature of the test surface can be achieved by means of induction hob 7. Fume extractor 8 with vent 9 captures the vapours generated as a result of evaporation and the pseudo-desublimation of AdBlue (pseudo-desublimation does not occur permanently). The vapours formed are sucked into a tube, which is cooled with Peltier element 10, so that their condensation takes place.

Experiments were carried out at different temperatures in order to know exactly whether and under what conditions pseudo-desublimation occurs for an AdBlue mixture. The results of these tests can be used to better understand the pseudo-desublimation phenomenon and to optimise the efficiency of the catalytic emission reduction process. Experiments were performed at different temperature thresholds using various injection times and varying the flow rate of AdBlue. For the analysis of the results, fast 240 fps movies were taken, both from top and side views of the test surface, using Trouble Shooter equipment produced by the Fastec Imaging Corporation Model TSHRCS.

After performing the experiments, the microtopography of a 1.5 mm by 2 mm region of the test surface, containing both a clean surface and solidified deposits, was mapped by aid of a Marsurf CWM 100 confocal microscope and interferometer. By evaluating the obtained microtopography parameters of the crystal deposits created on the test surface, the pseudo-desublimation process can be analysed.

## 5. Experimental Results

The images shown in Figure 15, Figure 16, Figure 17, Figure 18 and Figure 19 show extracts of the most relevant frames (selected frame numbers are indicated in figures in red) obtained for different combinations of temperature, injection time and injection flow rate. In these images, the darker features represent the test surface and the lighter features illustrate the solidified AdBlue droplets. In each figure, for constant experimental parameters regarding temperature, injection time and injection flow rate, the shown images present the time evolution of the phenomena.

The results of the experiments performed with the same flow rate of AdBlue injected on the heated surface showed the following cases:(a)Injection of AdBlue without vaporisation, Figure 15.

This situation occurs if the temperature of the combustion gases is not very high, for example, at start-up or idling when the maximum threshold does not exceed the vaporisation temperature. In this situation, the droplet jet encounters the walls of the tube and the SCR catalyst without instant vaporisation. Since the temperature is low in the SCR assembly, we have the worst case when the system does not catalytically convert the AdBlue solution, so we will have the highest level of nitrogen-based NOx.

(b)Injection of AdBlue in the case that the vaporisation process starts, Figure 16.

The AdBlue solution contains 32.5% urea solution and 67.5% demineralised water. Injected into the exhaust gas, AdBlue^®^ reduces harmful nitrogen oxide (NOx) emissions and complies with Euro 4, 5 and 6 pollution standards. According to ISO 22241 [39], Specifications for AdBlue, the boiling point is 100 °C. In other words, any temperature value at which the AdBlue solution is injected that exceeds this threshold will lead to the onset of vaporisation. The phenomenon is favoured by the lower droplets and flow rates, as well as the increased velocity.

(c)AdBlue injection in the case of pseudo-desublimation, Figure 17.

At temperatures between 130 and 140 °C for the tested surface in the case of low flow rates of the injected AdBlue, pseudo-desublimation is observed. This consists of the formation of a solid part adhering to the surface and of vapours which are entrained and incorporated into the microchannels of the SCR catalyst. From the experimental images in Figure 17, it can be seen that at the beginning of the process, the volume of the solid part Isocyanate (HCNO) is very small, but it adheres to the tested surface.

(d)Development of the pseudo-desublimation process.

At temperatures above 140 °C, a pronounced deposition on the heated surface with a lower evaporation rate was observed from the images in Figure 18. This may indicate that at this temperature, the evaporation process was accelerated and the AdBlue droplets have a higher tendency to deposit and crystallise on the heated surface.

(e)The appearance of the deflagration phenomenon of solid formations obtained after pseudo-desublimation.

In the temperature range between 200 and 300 °C, as shown in Figure 19, an instantaneous increase in the vaporisation rate of AdBlue was observed. Vaporisation occurs instantaneously from the first fraction of a second and manifests itself as a deflagration phenomenon. This result suggests that, at these particularly high temperatures, AdBlue vaporises so rapidly that the solid part no longer settles on the surface tested because it is separated into micrometric pieces by deflagration. As such, this phenomenon is favourable, as controlling the temperature inside the SCR can avoid the formation of solid deposits in the microchannels. We can conclude that lower temperatures favour the formation of crystals and the deposition of AdBlue, while higher temperatures lead to complete vaporisation and a reduction in residues.

These findings highlight the importance of temperature in the evaporation, reaction and deposition process of AdBlue, and temperature control and optimisation can play a crucial role in ensuring a uniform and efficient distribution of AdBlue in the SCR catalytic reduction system.

A detailed microscopic analysis of the deposits generated at a temperature of 230 °C, using the Marsurf CWM 100 confocal microscope, allows a quantitative evaluation of the crystals deposited on the test surface, as shown in Figure 20. The surface microtopography, shown in a 3D representation in Figure 20, indicates the distribution of solid deposits over the analysed sample. In the analysis, crystal height values, both surface and profile, were also taken into account for the studied sample.

In Figure 21, the profile of urea deposition on the same tested surface as Figure 20 is shown in 2D. A single profile taken from the surface microtopography and placed on the right side of Figure 21 shows the surface peaks and valleys that allow for further analysis, as described further.

Based on the obtained surface values found in Table 3, the profile roughness parameters Ra and Rz were calculated. The values obtained indicate that Ra = 31.122 μm (arithmetic mean of the absolute ordinate values within a sampling length of a profile) and Rz = 150.20 μm (arithmetic mean of the added distances between the five highest profile peaks and lowest profile valleys). It was found that the height of the crystalline deposit profile is influenced by the temperature at which urea decomposition took place. Thus, the temperature factor plays an important role in crystal formation and deposition on the surface as well as on the profile.

Sq surface roughness values were obtained using the Mahr MarSurf CWM 100 confocal microscope and interferometer (Figure 22). Sq is a surface texture parameter (ISO 25178 [41]) equivalent to the standard deviation of heights and is calculated as the root mean square value of the ordinate values within the defined area. Measurements were performed in the central region of the samples in 2 × 1.5 mm^2^ areas. Surface roughness was taken into account, so both directions were considered. The parameters were determined according to ISO 25178 using the default settings of the MountainsLab 8.1 software, which uses a predefined robust L-Gaussian filter of 0.8 mm.

These measurements allowed the surface roughness of the samples to be accurately assessed and the surface texture to be characterised in detail. The Sq values obtained provide important information about the variability in surface height and can be used to compare roughness levels between samples and to assess the effects of different methods or treatments on surface texture. By using appropriate tools and software, we can gain a deeper understanding of surface properties and their impact on the performance and functionality of components and systems. A sample of the profile of crystals deposited on the test plate can be observed in Figure 23. The highlighted regions of the profile indicate a clean surface region (right side of the image, between 0.86 mm and 1.21 mm) as well as a region with solidified deposits (left side of the image, between 0.07 mm and 0.81 mm). The measured parameters indicate for the 0.74 mm wide region with solidified deposits a maximum profile height of 138 µm and a mean height of 53.6 µm.

## 6. Conclusions

The present paper proves experimentally that in SCR systems the exhaust gas temperature is essential as a value for the occurrence of the pseudo-desublimation process. Knowing and accurately controlling the temperature of the exhaust gas–AdBlue mixture allows the SCR system software to ensure the injection process at optimal times to avoid the formation of solid deposits on the SCR manifold and inside the microchannels of this system.

The pseudo-desublimation of AdBlue microdroplets inside the microchannels of SCR systems is a process started by the conversion of the liquid AdBlue solution into microdroplets by injection. Once atomised, the high temperatures present in the SCR system transform the microdroplets into vapour that enters the SCR microchannels. Inside these microchannels, depending on temperature, the AdBlue vapour undergoes a desublimation process, which consists of a quasi-instantaneous transformation of the microdroplets into vapour or, at high temperatures, directly into solids. This process occurs due to the small size of the microchannels, their various shapes and the velocity of the fluid flowing through it.

The characteristics of the crystals formed and the rate of their growth depend on the working temperature of the gaseous medium, which can be controlled by efficient preheating methods immediately after engine start-up to the operating temperature. The use of such methods can help to avoid solid depositions on SCR system components, which have a direct impact on SCR catalyst performance and durability.

## Figures and Tables

**Figure 1 micromachines-14-01807-f001:**
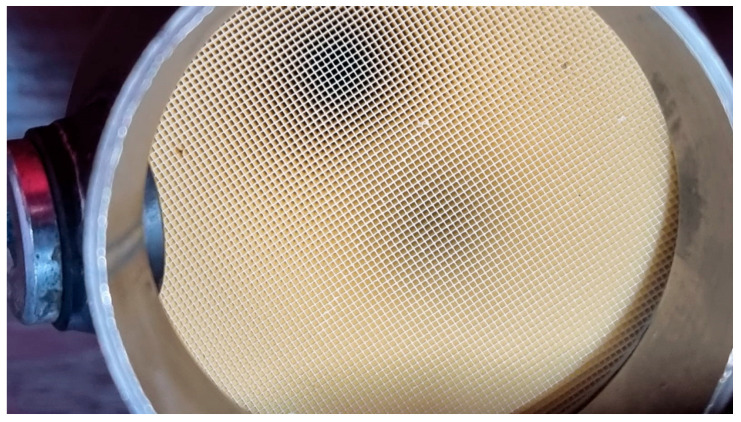
Example of microchannels of a SCR system.

**Figure 2 micromachines-14-01807-f002:**
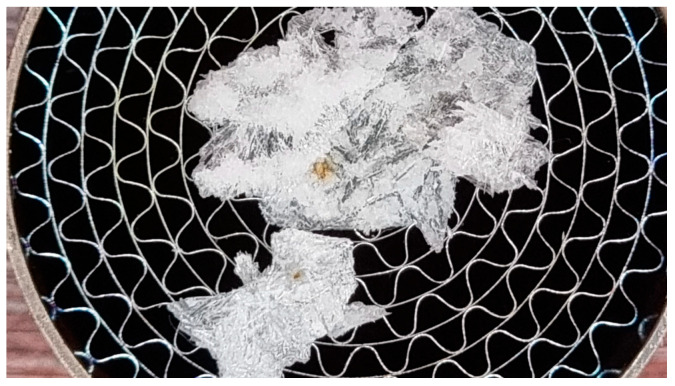
Pseudo-desublimation effect in SCR systems.

**Figure 3 micromachines-14-01807-f003:**
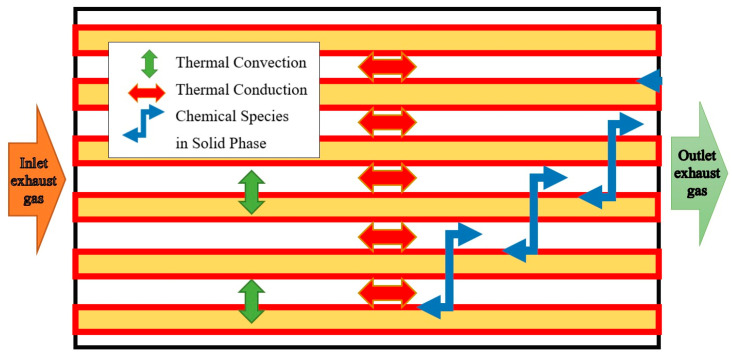
Thermal phenomena occurring in microchannels with circular geometry.

**Figure 4 micromachines-14-01807-f004:**
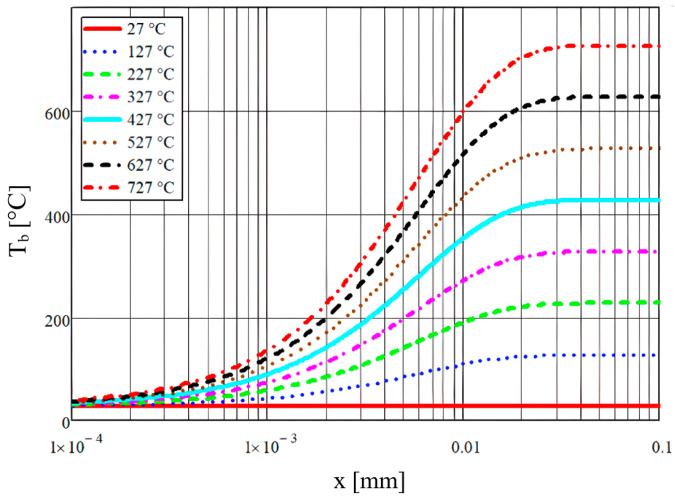
Temperature variations over the microchannel length for various gas temperature values (marked in the graph legend).

**Figure 5 micromachines-14-01807-f005:**
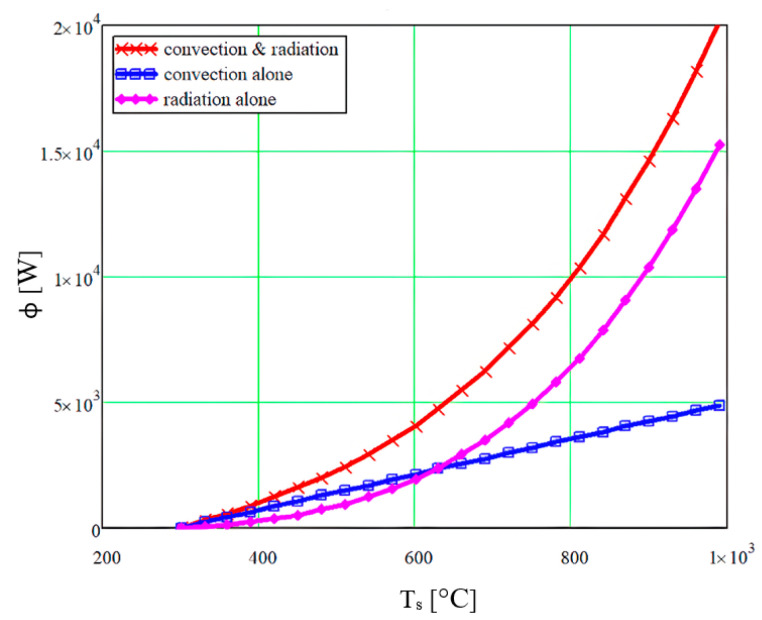
Variation in heat transfer with exhaust gas temperature for convection, radiation and both in a SCR.

**Figure 6 micromachines-14-01807-f006:**
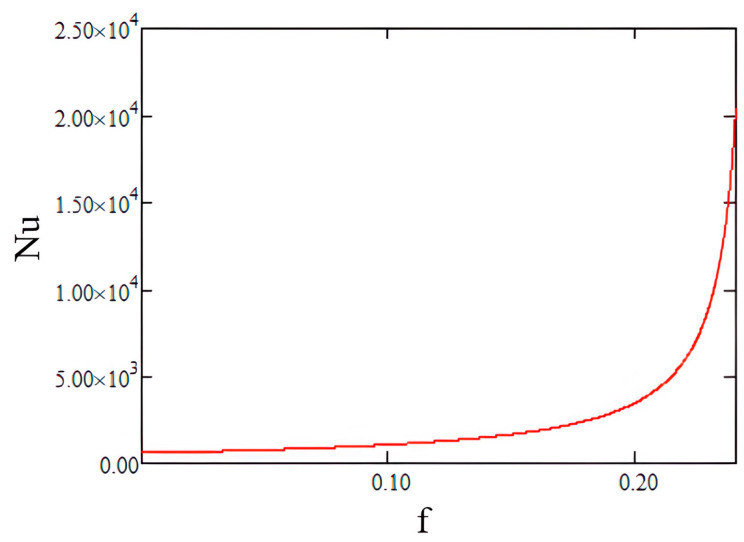
Variation of Nusselt number as a function of friction coefficient.

**Figure 7 micromachines-14-01807-f007:**
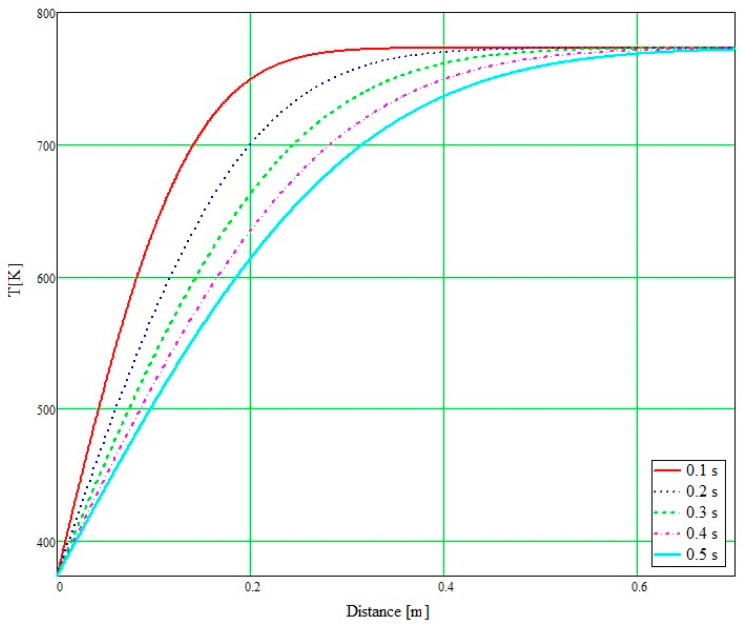
Transient temperature variation along the SCR.

**Figure 8 micromachines-14-01807-f008:**
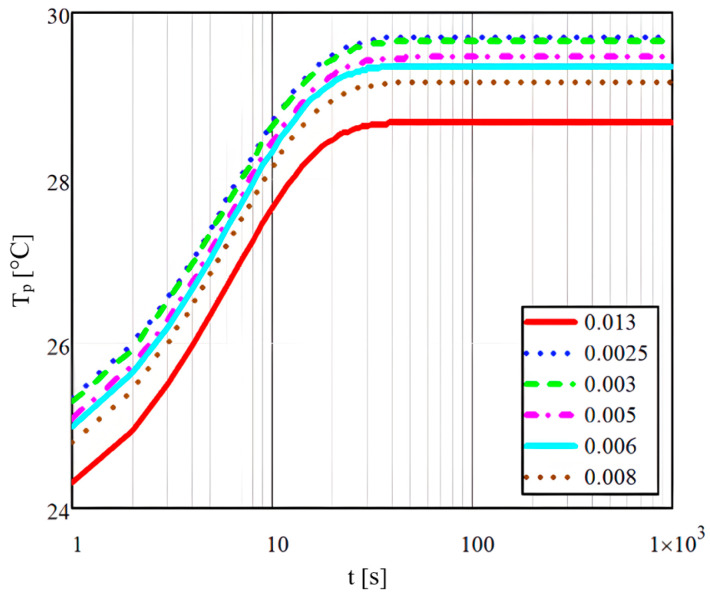
Transient heat flow at different microchannel wall thicknesses.

**Figure 9 micromachines-14-01807-f009:**
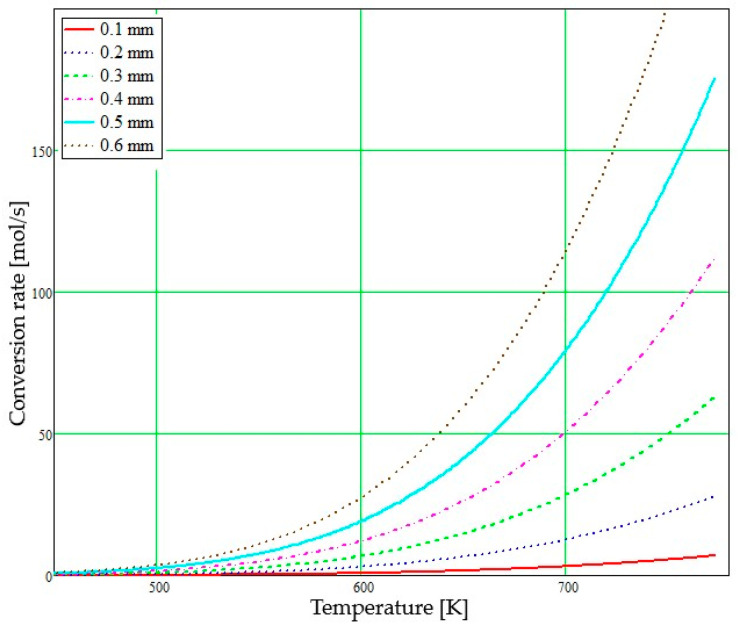
Conversion rate of exhaust gas through microchannels with different diameter sizes.

**Figure 10 micromachines-14-01807-f010:**
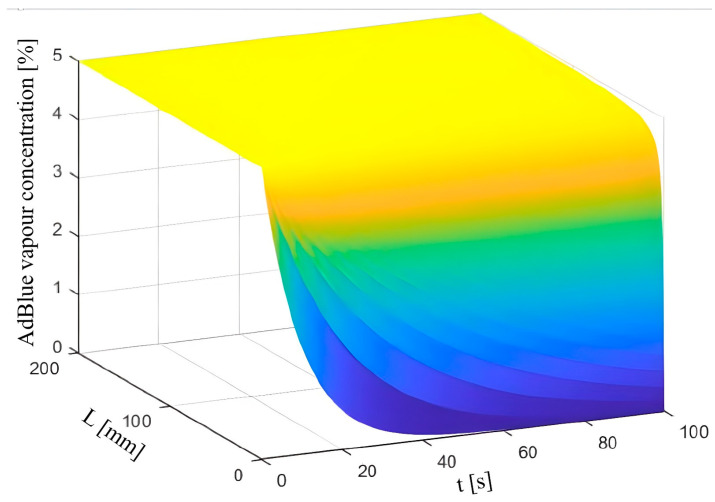
Aerosol concentration evolution as a function of microchannel length and time.

**Figure 11 micromachines-14-01807-f011:**
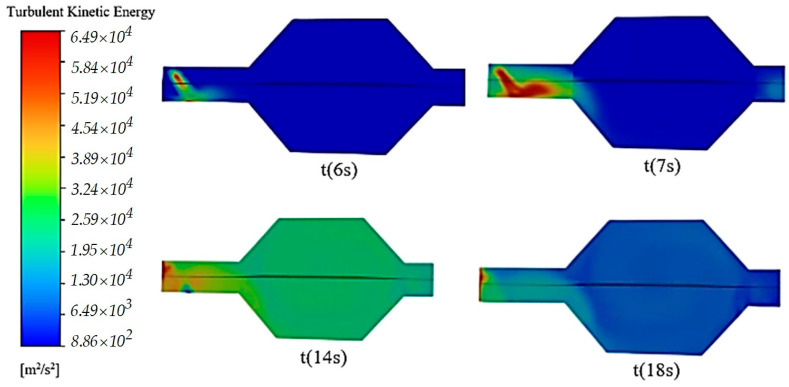
AdBlue vapour flow simulation in SCR system.

**Figure 12 micromachines-14-01807-f012:**
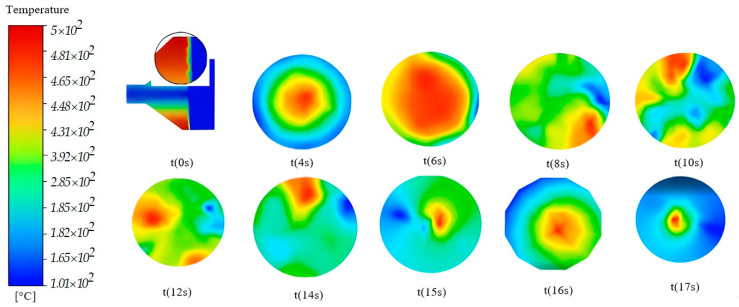
Temperature variation along an SCR microchannel traversed by the mixture.

**Figure 13 micromachines-14-01807-f013:**
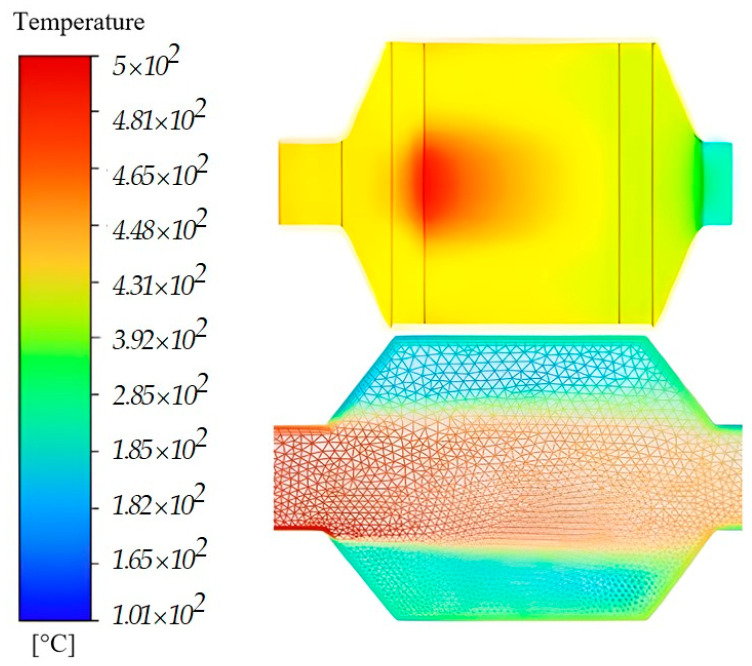
Temperature field in an SCR system.

**Figure 14 micromachines-14-01807-f014:**
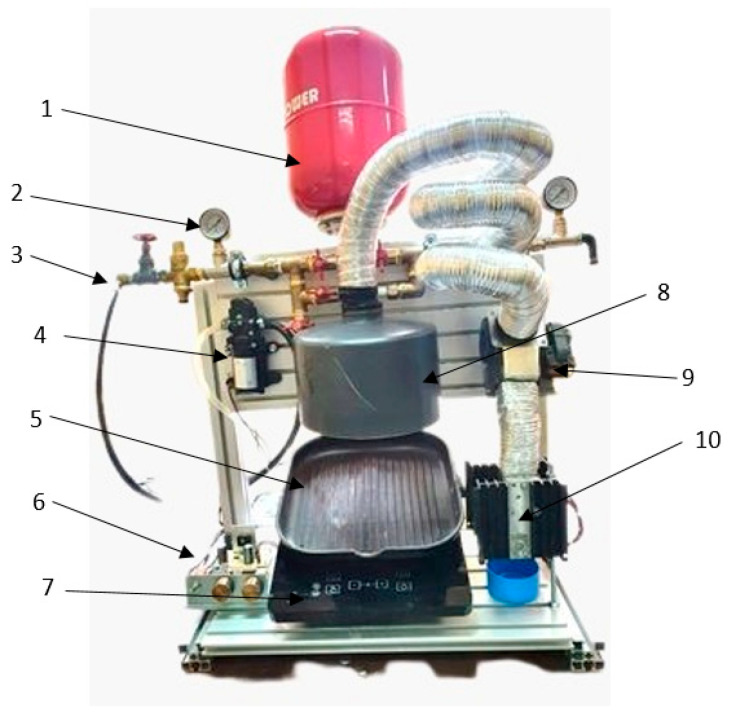
Experimental set-up for the study of AdBlue pseudo-desublimation.

**Figure 15 micromachines-14-01807-f015:**
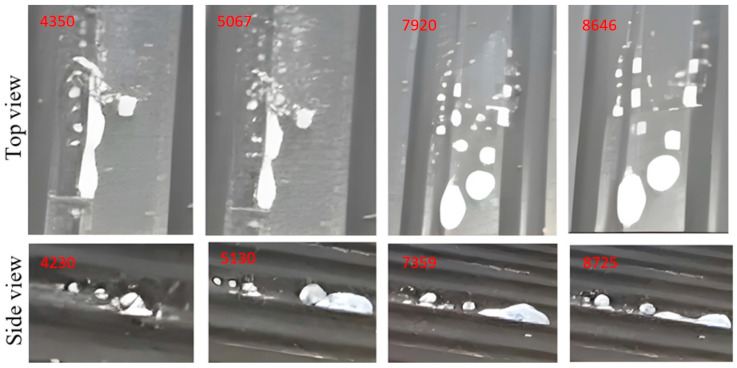
Injection of AdBlue without vaporisation.

**Figure 16 micromachines-14-01807-f016:**
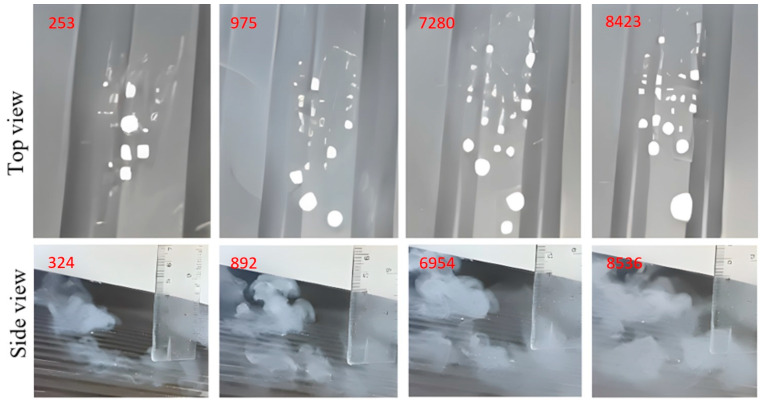
AdBlue injection when vaporisation starts.

**Figure 17 micromachines-14-01807-f017:**
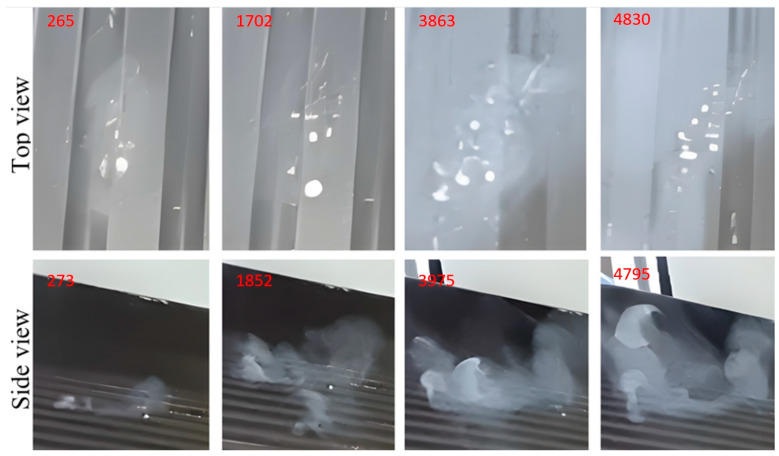
AdBlue injection in case of pseudo-desublimation.

**Figure 18 micromachines-14-01807-f018:**
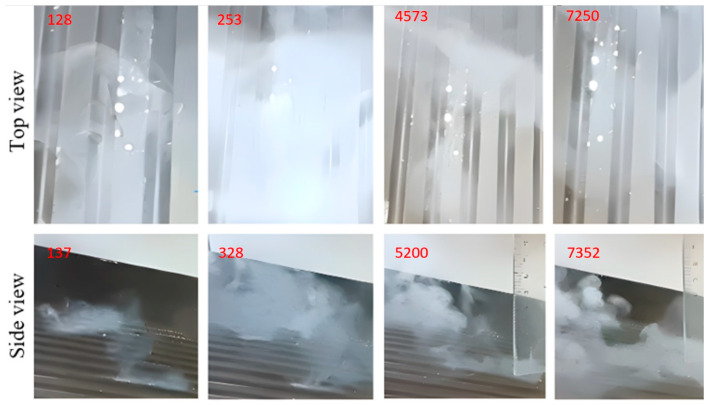
Development of the pseudo-desublimation process.

**Figure 19 micromachines-14-01807-f019:**
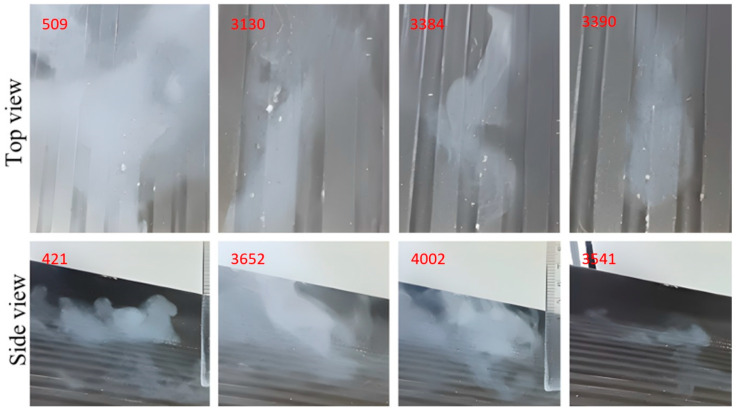
Occurrence of deflagration phenomenon of solid formations obtained after pseudo-desublimation.

**Figure 20 micromachines-14-01807-f020:**
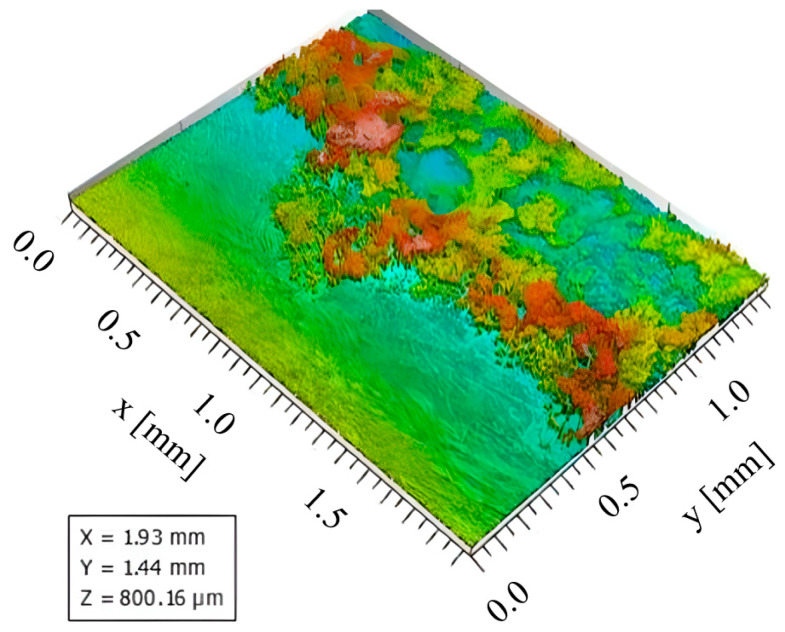
Surface microtopography of pseudo-desublimation deposits.

**Figure 21 micromachines-14-01807-f021:**
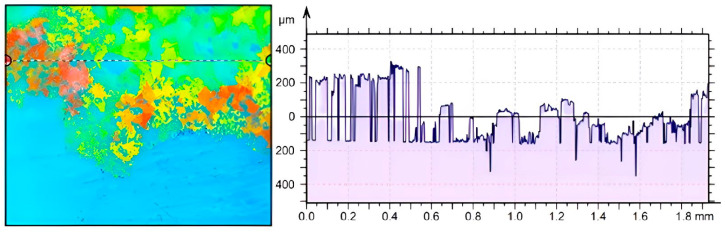
Area of interest on the test plate and profiles of crystals deposited as a result of the pseudo-desublimation process.

**Figure 22 micromachines-14-01807-f022:**
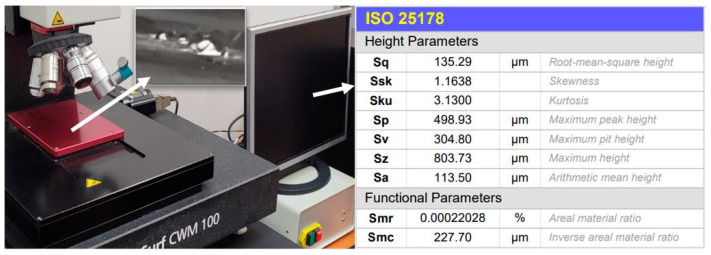
Surface microtopography measurement.

**Figure 23 micromachines-14-01807-f023:**
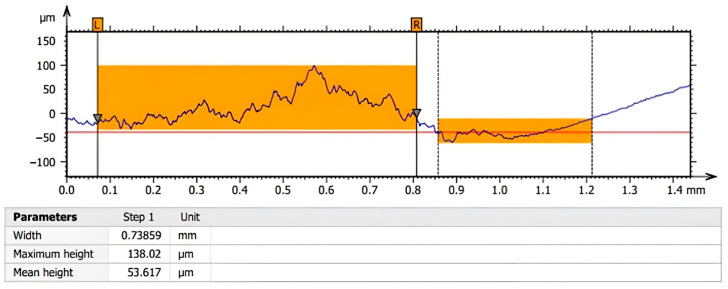
Profile of crystals deposited on the test plate observed using the CWM 100 confocal microscope.

**Table 1 micromachines-14-01807-t001:** The amount of pollutant according to European requirements for engines up to 35 Kw diesel and petrol.

Pollutant	Unit	Euro 7	Euro 6d
NO_x_	mg/km	60	60/80
PM	mg/km	4.5	4.5/4.5
PN	-	$6\times 10^{11}$	$6\times 10^{11}$ $/6\times 10^{11}$
CO	mg/km	500	1000/500
THC	mg/km	100	100/-
NMHC	mg/km	68	68/-

**Table 2 micromachines-14-01807-t002:** SCR parameters.

Parameter	Value
Catalyst honeycomb total size	100 mm×50 mm×200 mm
Catalyst honeycomb microchannel diameter	0.8 mm
Catalyst honeycomb microchannel length	200 mm
Composition of exhaust gases	NO_x_ = 50 ppm, NH_3_ = 100 ppm, CO_2_ = 10%
Exhaust gas temperature	400 °C
Ambient temperature	25 °C
Exhaust gas pressure	$1.32\times 10^{5}\;\mathrm{Pa}$
Exhaust gas velocity	10 m/s
Catalyst thermal conductivity	0.5 W/mK
Heat capacity of catalyst	900 J/kgK
Catalyst density	800 kg/m^3^
Initial catalyst temperature	200 °C

**Table 3 micromachines-14-01807-t003:** Surface amplitude parameters (ISO 4287 [40]).

Amplitude Parameter	Unit	Values
Rz	μm	150.20
Ra	μm	31.122

## Data Availability

Some or all data, models or code generated or used during the study are available from the corresponding author by request.

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
