# Peer review of "Pseudo-Desublimation of AdBlue Microdroplets through Selective Catalytic Reduction System Microchannels and Surfaces"

_micromachines, 2023, doi:10.3390/mi14091807_

Round 1

Reviewer 1 Report

In this study, the authors have conducted an investigation of the pseudo-desublimation process of AdBlue microdroplets within the microchannels and surfaces of selective catalytic reduction (SCR) systems. The study analyzes the influence of heat flux values on the heat transfer of AdBlue injection, considering the structure of the microchannels and the overall configuration of the SCR installation. Additionally, the analysis explores the exhaust gas conversion rate versus temperature and microchannel diameter. There are several topics in the paper, however, that require improvement before it can be considered for publication in the Micromachines Journal:

1.     The abstract should highlight the key findings of the study, which are currently missing.

2.     In the introduction, the authors should provide a clear rationale for the selection of different shapes of microchannels and their relevance to the study. It is crucial to explain why these specific shapes have been chosen, where they can be commonly found, and what their potential applications are. Given the extensive existing research in this field, it is of utmost importance to highlight the novelty and potential significance of this numerical study in the manuscript. To accomplish this, it would be beneficial to reference relevant studies such as (DOI: 10.1039/C5NR05828G; DOI: 10.1016/j.electacta.2022.141175; DOI: 10.1021/acs.jpcc.6b08588). These references can provide additional support and context for the research, reinforcing its originality and relevance within the field.

3.     The introduction lacks clarity in providing a brief background and motivation for the study. It would be beneficial to explain the significance and potential applications of the pseudo-desublimation process in AdBlue injection within SCR systems.

4.     The methodology section needs further elaboration. Details regarding the experimental setup, including the high-speed camera technology and the CWM 100 confocal microscope, should be provided to ensure reproducibility and clarity.

5.     What specific heat flux values were considered in the analysis, and how do they affect the heat transfer of AdBlue injection in the SCR system?

6.     What are the practical implications of understanding the pseudo-desublimation process in AdBlue injection within SCR systems? Can this knowledge contribute to improving the efficiency or performance of SCR systems? Please explain this issue in the text.

7.     The authors should clarify the observed trends in the conversion rate of exhaust gases concerning temperature and microchannel diameter. Are there any specific relationships or patterns observed?

8.     The conclusion should summarize the main findings and their significance in a concise manner. Additionally, future research directions or recommendations for improving the efficiency or performance of the SCR systems should be included.

9.     Please edit the language, fix typos, and correct grammatical errors. The layout of references should be also thoroughly revised.

My comment has been mentioned in the suggestions for authors.

Reviewer 2 Report

This manuscript may contain useful information about modeling and experimental work performed on urea injection in an automotive selective catalytic reduction system, but the writing and presentation are poor enough that I am uncertain.  I am recommending rejection, based on the lack of care evident in the submitted manuscript.  However, if the editor chooses to proceed with publication, I recommend that the authors address the following numerous issues.

The Introduction is not clear.  “The new cold emissions test will mean big increases, as petrol engines had a limit of 60 mg/km for NOx emissions, while diesel engines had a limit of 80 mg/km.”  Big increases of what?  The whole paragraph should be clarified.

Line 58 states an NH3 emission limit of 10 ppm is shown in Table 1, but no such limit is shown in this table.  Please correct either the table or the text.  Also in Table 1, a pollutant called “PH10” with a limit of “6 x 1011” is shown.  This should be explained in the text.  Now that I have seen Table 2, I suspect that there is a missing superscript here and these numbers should read “6 x 10^11”, but it all needs to be corrected.  In general, Table 1 is poorly explained.  While I believe THC stands for “total hydrocarbons”, not everyone who reads Micromachines will be familiar with these and other acronyms.  Please help the reader by defining/writing acronyms out at first use.

The paragraph in lines 87-96 contains several redundant statements and should be rewritten much more concisely.

In several instances in the manuscript (lines 67, 131, 136, 428, 436, and 494), the process is called “pseudo-sublimation” instead of “pseudo-desublimation”.  I presume this wording is incorrect, or is there both a sublimation and desublimation process going on?  Please correct the typographical errors or explain the difference between these two processes.

In line 13, “in” should be subscripted in “Cin”.

Equation 2 contains either a typographical error or a font-rendering problem.  There is a “?” in the middle of the equation that I presume should be an “inverse delta” or “del” or other mathematical symbol to make the transport equation correct.  Equations 3, 4, 13, 15, 17, and 20 also contain unknown symbols (they appear to be Chinese) and Equation 4 has another question mark that all must be corrected.

In line 145, the symbol for “rho” does not appear to be the same one used in Equation 2.

In Equation 6 and line 166, “p” should be subscripted in “Cp”, as well as in line 183 and Equation 11.

Table 2 does not contain appropriate superscripting and subscripting.  This is vital in general, but especially for the exhaust gas pressure, which should be “1.32 x 10^5 Pa”, not “1.32 x 105 Pa” as shown.

Please report numbers to an appropriate level of engineering significant figures.  For example, in line223, the Nusselt number is reported to 5 significant figures.  I believe the accuracy of the model (and the correlations that go into it) do not justify this level of precision.  Both of the Figure 22’s and Table 3 (and the values in the text reported from these data sources) should be similarly scrutinized.

Figure 7 is fairly meaningless without further explanation.  What are all of the lines?  What transient is this representing?  The x-axis shows time, indicating a static measurement of time which would not allow for a transient.  Is the x-axis supposed to be distance and the lines show the change of T with increasing time?  Something is either seriously wrong with this graph or it requires much more explanation than has been presented.

Line 247 appears to reference a paper that is given the number 32 in the References.  The paper should be referenced by this number and if the authors’ names are used in the text, it must be shown as Mihai et al.

The variables in Equation 18 must be defined.  What is Tp?  What is “delta subscript t”?  What is C? 

Equation 21 is mislabeled as equation 20.  What is E in this equation?  Please explain how this equation has physical meaning.  It is not shown as a function of time, yet Figure 10 indicates it is a function of time.  Is the equation implicit with respect to time?  If so, how?  Why is Figure 10 only plotted for 20 mm?  Is this a start-up calculation?

The description of Figure 12 does not make sense.  First, it is stated that “AdBlue injection takes place at t(8s)”, but then “The first image corresponds to the initial time t(0s), which is associated with the period when the droplet jet encounters the microchannels of the SCR system.”  What droplet jet?  The injection has not taken place until 8 s, so how can there be any droplets?

The Experimental portion of the manuscript is poorly presented.  There should be a separate “Experimental” or “Methods” section of the manuscript.

Figures 15 and 16 need to be described such that someone who is unfamiliar with the process (i.e., the average reader) can understand them.  What are the light and dark features?  Are the light patches droplets of AdBlue?  Are they solids?  Is time increasing left to right?  What are the time steps?  Just stating that the camera takes images at 240 frames per second does not mean anything unless they are consecutive images, which is not at all apparent.  Do your readers the courtesy of writing with sufficient clarity to make your work understandable to someone who has not performed it.  Many of the same questions apply to Figures 17-19, although these are a little clearer as the vapor formation is more apparent.

Table 3 includes many parameters (presumably described in ISO 4287) that are not discussed in the manuscript, other than Ra and Rz.  If you do not discuss them, do not include them.

There are two Figure 22’s in the manuscript.

Lines 504-505 state that “All authors have read and agreed to the published version of the manuscript.”  Taking this affirmation as true, the authors have done a careless job of proofreading, which if similar care was taken in their modeling and experimental work, makes me question the reliability of their results.

Minor typographical issue:
In line 69, the word “which” apparently should be deleted.

The English quality could be improved, but it is generally sufficient.  What is lacking is appropriate effort by the authors to present their work clearly.

Reviewer 3 Report

The subject investigated in the paper is very specific and might be of interest to  a limited number of readers of Micromachines, yet the paper is easy to follow and self-contained,  This Reviewer deems the paper publishable in its current form. The only chnange that must be made is using parentheses instead of brackets to enclose units both in the text and in the plots.

Author Response

Dear reviewer,

We are very grateful for the reviews and for the provided suggestions.

As suggested, in the revised version of the manuscript, parantheses were employed for units in text instead of brackets.

Reviewer 4 Report

1. There are errors and typos in the text of the article, carefully re-read the article and correct it.

2. Table 2 shows the parameters of the catalyst honeycomb size - specify the total size or one cell, specify the size separately and the total.

3. Add a Nomenclature with dimensions of all components.

4. Carefully check all the equations, in some the dimensions of the given values do not match.

5. On page 7, figure 4 (indicated that 2)

6. Why is there such a frequency of lines in Figure 7?

7. In the experimental results section to Figures 15-19, sign what is indicated (probably the time).

8. To describe the results in more detail in Figure 20-21

9. There is a typo in formula 20

Moderate editing of English language required

Round 2

Reviewer 1 Report

I am satisfied with the revised version of the manuscript.

The English are ok. The text is very well understandable.

Author Response

Dear reviewer

We are very grateful for your review.

Reviewer 4 Report

The authors responded to the comments as best they could

Minor editing of English language required